# Three-body correlations in nonlinear response of correlated quantum liquid

Tokuro Hata [1✉], Yoshimichi Teratani [2], Tomonori Arakawa[1,3], Sanghyun Lee [1], Meydi Ferrier[1,4], Richard Deblock[4], Rui Sakano [5], Akira Oguri[2,6] & Kensuke Kobayashi [1,7,8✉]

Behavior of quantum liquids is a fascinating topic in physics. Even in a strongly correlated case, the linear response of a given system to an external field is described by the fluctuation-dissipation relations based on the two-body correlations in the equilibrium. However, to explore nonlinear non-equilibrium behaviors of the system beyond this well-established regime, the role of higher order correlations starting from the three-body correlations must be revealed. In this work, we experimentally investigate a controllable quantum liquid realized in a Kondo-correlated quantum dot and prove the relevance of the three-body correlations in the nonlinear conductance at finite magnetic field, which validates the recent Fermi liquid theory extended to the non-equilibrium regime.

[1] Graduate School of Science, Osaka University, Toyonaka, Osaka, Japan. [2] Department of Physics, Osaka City University, Osaka, Japan. [3] Center for Spintronics Research Network, Osaka University, Toyonaka, Osaka, Japan. [4] Université Paris-Saclay, CNRS, Laboratoire de Physique des Solides, Orsay, France. [5] The Institute for Solid State Physics, The University of Tokyo, Chiba, Japan. [6] Nambu Yoichiro Institute of Theoretical and Experimental Physics, Osaka City University, Osaka, Japan. [7] Institute for Physics of Intelligence and Department of Physics, The University of Tokyo, Bunkyo-ku, Tokyo, Japan. [8] Trans-scale Quantum Science Institute, The University of Tokyo, Bunkyo-ku, Tokyo, Japan. ✉email: hata@phys.titech.ac.jp; kensuke@phys.s.u-tokyo.ac.jp

Understanding the properties of correlated quantum liquids is a fundamental issue of condensed matter physics. Even in a strongly correlated case, fascinatingly, we can tell that the equilibrium fluctuations of the system govern its linear response to an external field, relying on the fluctuation-dissipation relations based on the two-body correlations. Going beyond this well-established regime, the three-body correlations are known to be of importance for van der Waals force[1], the three-body force in nuclei[2], the Efimov state[3,4], the ring exchange interaction in solid $^3$He[5,6], and frustrated spin systems[7].

A quantum dot (QD) shows the Kondo effect, when a localized spin in the QD is coupled with conduction electrons in the reservoirs to form a spin-singlet state[8–11]. The QD in the Kondo regime is an ideal realization of a strongly correlated quantum liquid with high controllability and accessibility to the spin degree of freedom. The essential physics can be captured by the Anderson impurity model, which describes the single impurity state with spin $\sigma(=\uparrow, \downarrow$ or $\pm 1)$ and energy $\varepsilon_\sigma$ coupled to the two metallic reservoirs. The Fermi liquid theory for the Anderson model, which has been developed phenomenologically[12] and microscopically[13–17], can explain the low-energy physics of this highly correlated quantum system as the properties of an ensemble of quasi-particles interacting with each other via residual interaction.

At zero magnetic field, namely when the time-reversal symmetry (TRS) is present, the linear response of the Kondo state can be explained in terms of only a few parameters; the phase shift $\delta_\sigma$ of the electron passing through the impurity state, and the two-body correlations ($\chi_{\sigma_1\sigma_2}$) between spin $\sigma_1$ and spin $\sigma_2$. Here, the two-body correlation is identical to the conventional linear susceptibility, which is defined as the differential of the free energy $\Omega$ of the total system regarding the energies of two electrons, $\chi_{\sigma_1\sigma_2} \equiv -\partial^2\Omega/\partial\varepsilon_{\sigma_1}\partial\varepsilon_{\sigma_2}$ (see Supplementary Note 1). These parameters determine the two crucial physical quantities of the Kondo problem: the Kondo temperature $k_BT_K \equiv 1/(4\chi_{\uparrow\uparrow})$ and the Wilson ratio $R \equiv 1 - \chi_{\uparrow\downarrow}/\chi_{\uparrow\uparrow}$[12,13].

To go beyond this well-established regime, nonlinear susceptibility, that is, three-body correlation $\chi_{\sigma_1\sigma_2\sigma_3}$ comes into the game (see Fig. 1a). Similarly as above, it is defined as $\chi_{\sigma_1\sigma_2\sigma_3} \equiv -\partial^3\Omega/\partial\varepsilon_{\sigma_1}\partial\varepsilon_{\sigma_2}\partial\varepsilon_{\sigma_3}$. To give an intuitive picture, consider a situation where three electrons with spin $\sigma_1$, $\sigma_2$, and $\sigma_3$ pass through the QD level in sequence as schematically shown in Fig. 1b. The electron with $\sigma_1$ first occupies and leaves the level, and then the second electron with $\sigma_2$ passes through it. Finally,

the third electron with $\sigma_3$ occupies it. This process describes that the spin in the QD level fluctuates from $\sigma_1$ to $\sigma_2$ to $\sigma_3$ (from $\uparrow$ to $\uparrow$ to $\downarrow$, in this specific case shown in Fig. 1b) and that the spin of the first electron affects that of the third electron. The correlation among $\sigma_1$ and $\sigma_3$ via $\sigma_2$, or three-spin exchange process, gives the most naive picture of the three-body correlations, which constitute the leading term of the nonlinear response of the system as characterized by $\chi_{\sigma_1\sigma_2\sigma_3}$ ($\chi_{\uparrow\uparrow\downarrow}$ in this case).

In this work, using a carbon nanotube QD in the $SU(2)$ Kondo regime, we experimentally prove that the three-body correlations $\chi_{\sigma_1\sigma_2\sigma_3}$ indeed contribute to the nonlinear conductance. Thanks to the quality of our sample, where the Kondo effect in the unitary limit is achieved, we quantitatively measure the three-body correlations, in perfect agreement with recent results of the Fermi liquid theory[18–23], and verify their role in the non-equilibrium regime. In particular, we demonstrate their importance when TRS is broken, and solve a long-standing puzzle of the Kondo system under the magnetic field[19]. The demonstrated method to relate three-body correlations and non-equilibrium transport opens up a way for further investigation of the dynamics of quantum many-body systems.

## Results

**Nonlinear conductance in Kondo regime.** We first outline the basic idea of the present study. We examine the differential conductance ($dI/dV$) of the Kondo-correlated QD with the following expansion in terms of the bias voltage ($V$)[24–30]:

$$\frac{dI}{dV} = G_0 - \alpha_V\left(\frac{eV}{k_BT_K}\right)^2 + \cdots, \quad (1)$$

at $T \ll T_K$. Here $G_0$, $e$, and $k_B$ are the zero-bias conductance, elementary charge, and the Boltzmann constant, respectively. The coefficient $\alpha_V$ of the order $V^2$ consists of two- and three-body terms, $W_2$ and $W_3$, respectively. As discussed later, $W_2$ is defined using $\delta_\sigma$ and $\chi_{\sigma_1\sigma_2}$ (and hence $R$) [see Eq. (2)], which was previously addressed and established experimentally[31–35]. In this work, we newly focus on $W_3$, which is the function of $\delta_\sigma$, $\chi_{\uparrow\uparrow}$, $\chi_{\uparrow\uparrow\uparrow}$, and $\chi_{\uparrow\uparrow\downarrow}$ for specific cases [see Eq. (7), Methods, and Supplementary Note 1][18–23]. In the analysis, we rely on the fact that $G_0 = e^2/h \times (\sin^2\delta_\uparrow + \sin^2\delta_\downarrow)$ in the Kondo effect in the unitary limit. This treatment is valid in this experimental work as we have achieved the unitary limit in our QD (see Fig. 2a)[35,36].

Importantly, when the on-site Coulomb interaction ($U$) in the QD is zero, a complete analytical form for Eq. (1) is obtained by

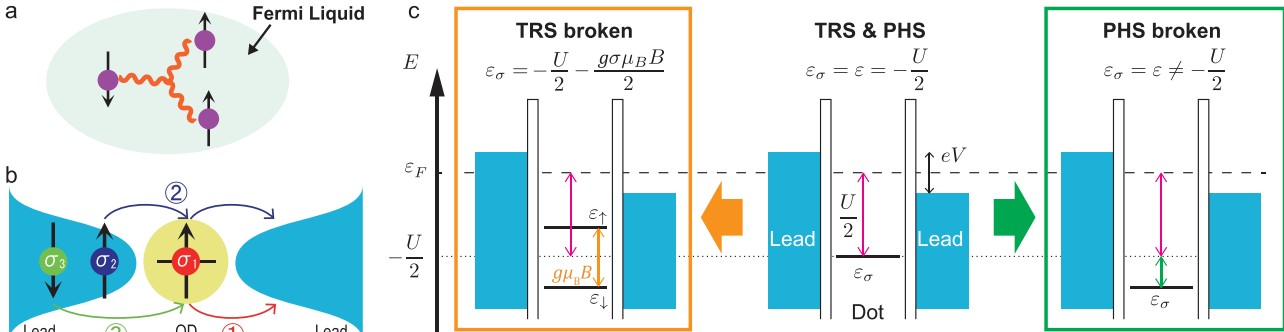

**Fig. 1 Three-body correlations and symmetry breaking in a quantum dot in the Kondo regime. a** Schematic view of the correlations between the three electrons accounting for the Fermi liquid correction. **b** Schematic view of the three-body correlation $\chi_{\uparrow\uparrow\downarrow}$. Electrons fluctuate between the quantum dot (QD) and the leads in the equilibrium and all those processes are summed up in $\chi_{\uparrow\uparrow\downarrow}$. **c** (center) QD with the time-reversal symmetry (TRS) and particle-hole symmetry (PHS). The energy level is $\varepsilon_\sigma = -U/2$. The two-body correlations $\chi_{\sigma_1\sigma_2}$ are finite, while the three-body correlations are quenched. (left) QD with broken TRS, where the two spins are separated by $g\mu_B B$. (right) QD with broken PHS, where the energy level is $\varepsilon \neq -U/2$. Experimentally, the gate voltage is applied to the QD in order to tune the energy level. In these broken symmetry cases, the three-body correlations $\chi_{\sigma_1\sigma_2\sigma_3}$ are finite, which we detect in this paper.

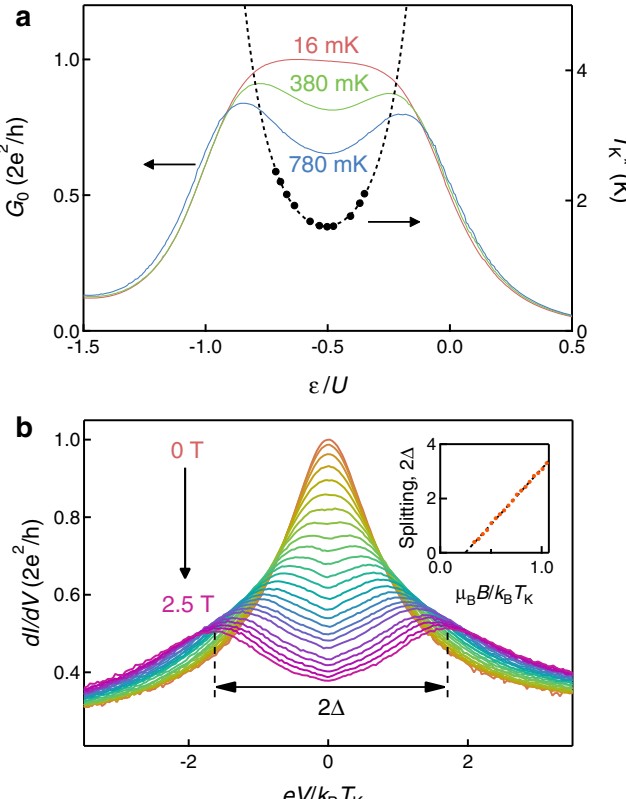

**Fig. 2 Energy level and magnetic-field dependence of the Kondo state.**
**a** $\varepsilon/U$ dependence of $G_0$ for three temperatures (16, 380, and 780mK). The experimental $T_K^*$ shown on the right axis is fitted with the formula shown in ref. [35]. **b** Differential conductance at the PHS point as a function of $eV/k_B T_K$ for different magnetic fields from 0 to 2.5T with 0.1T steps. Here, the lowest field 0.1T is defined as 0T (see text). (inset) $2\Delta$, which is the width of Zeeman splitting, as a function of $\mu_B B/k_B T_K$.

replacing $k_B T_K$ with $2\gamma_0$ (see Methods and Supplementary Note 1). Here, $2\gamma_0$ virtually corresponds to the half-width of a resonance peak. Note that the three-body correlations for the same spins ($\chi_{\uparrow\uparrow\uparrow}$ and $\chi_{\downarrow\downarrow\downarrow}$) are finite even in this $U=0$ case due to the Pauli exclusion principle. On the other hand, only for the $U\neq0$ case, the Kondo effect makes other components such as $\chi_{\uparrow\uparrow\downarrow}$ finite. Thus, the experimental observation of the deviation from the $U=0$ case, which we refer as the free particle (FP) model later, directly tells us the relevance of those correlations in the nonlinear Kondo regime. In addition, recently, the expression of the coefficient $\alpha_V$ in the interacting case was theoretically obtained by expanding the Fermi liquid theory to the non-equilibrium regime[18–23]. Our present work, relying on a precise experiment, aims to perform an accurate test of this theory.

To this end, we experimentally examine $\alpha_V$ when TRS or the particle-hole symmetry (PHS) is broken. In the presence of the magnetic filed ($B$), a single electron energy with spin $\sigma$ in the QD is expressed as $\varepsilon_\sigma = \varepsilon - g\sigma\mu_B B/2$, where $\mu_B$ and $g\approx2$ are the Bohr magneton and the $g$−factor in the nanotube, respectively. When both TRS and PHS hold, the energy level is located at $\varepsilon_\sigma = -U/2$ (center of Fig. 1c), the situation that many experimental works have addressed[31–35]. In this symmetrical point, the two-body correlations are finite, while the three-body correlations are quenched due to the symmetry. We systematically tune $\varepsilon_\sigma$ to investigate the behavior of $\alpha_V$ by varying either $B$ or the gate voltage to vary $\varepsilon$. As is known in the Kondo physics, the magnetic field splits the degenerate level by a quantity $g\mu_B B$, which breaks TRS (see the left panel of Fig. 1c). The gate voltage, on the other

hand, moves $\varepsilon$ from $-U/2$ and breaks PHS (see the right panel of Fig. 1c)[18–21].

**Sample and measurement setup.** We perform conductance measurement for a carbon nanotube QD connected to two Pd/Al electrodes in the dilution fridge, using a standard low-frequency lock-in technique. We focus on the conventional $SU(2)$ Kondo state previously reported in refs. [35–37]. The details are presented in those references and in Supplementary Note 4. The shape of the Coulomb diamond and the shot noise measurement experimentally determines whether the symmetry of the Kondo state is $SU(2)$ or $SU(4)$[35–37]. The base temperature of the dilution fridge is 16 mK. A small magnetic field 0.1 T, which is sufficiently small compared to the Kondo temperature ($T_K = 1.6$ K at the PHS point), is always applied to suppress the superconductivity of the electrodes. In this paper, for simplicity, we refer to this lowest field 0.1 T as zero field (0 T), except when we mark experimental points at $B = 0.1$ T in the figures.

**Basic characteristics of the Kondo state.** Figure 2a shows the zero-bias conductance $G_0$ as a function of $\varepsilon/U$ at zero field for three different temperatures. In this paper, we define $T_K$ as the Kondo temperature at the TRS and PHS point ($B = 0$T and $\varepsilon/U = -0.5$). We also define $T_K^*(\varepsilon, B)$ as the Kondo temperature for general cases. By definition, $T_K^*(\varepsilon/U = -0.5, B = 0$ T$) \equiv T_K = 1.6$ K. The black points in Fig. 2a are the Kondo temperatures $T_K^*(\varepsilon, B = 0$ T$)$, which we evaluate by analyzing the temperature dependence of $G_0$[35]. We obtain $U = 6.4 \pm 0.8$ meV and $\Gamma = 1.9 \pm 0.2$meV by fitting the gate voltage dependence of $T_K^*(\varepsilon, B = 0$ T$)$ (dashed curve), where $\Gamma$ is the width of energy levels due to coupling between the electron in the QD and the lead electrons[35]. Together with $U/\Gamma = 3.4 \pm 0.6$, $R = 1.95 \pm 0.05$, which we independently determined from the shot noise experiment[35], indicates that the Kondo state is very close to the limit of the strong correlation[25]. As the two-body correlation $W_2$ is defined[20–23] by

$$W_2 = -\left[1 + 5(R-1)^2\right]\sum_\sigma \frac{\cos 2\delta_\sigma}{2}. \quad (2)$$

$W_2 = 5.5$ at $R = 1.95$ and the symmetric point ($\delta_\sigma = \pi/2$).

Figure 2b shows the differential conductance obtained at 16 mK and the PHS point ($\varepsilon/U = -0.5$) as a function of $eV/k_B T_K$ for different $B$ from 0 to 2.5 T. When $B$ increases, the Kondo resonance gets split, and its amplitude is reduced. Inset in Fig. 2b shows the Zeeman splitting $2\Delta$ as a function of $\mu_B B/k_B T_K$. The linear fitting shows that the $g$-factor is $2.0 \pm 0.05$ just as expected for carbon nanotube and the splitting seems to start at $\mu_B B/k_B T_K = 0.23 \pm 0.02$ ($B = 0.6$T). Although this splitting has served as a hallmark of the Kondo effect[8–11], the microscopic mechanism has been theoretically revealed only recently[19–23].

**Two- and three-body correlations in TRS broken case.** We analyze the magnetic field dependence of the Kondo peak at the PHS point with Eq. (1) (the analysis procedure is detailed in the Supplementary Note 3). At a finite magnetic field, the phase shift relates to the induced magnetization of the electron inside the QD ($m_d$) such that

$$m_d = \frac{\delta_\uparrow - \delta_\downarrow}{\pi}. \quad (3)$$

Then, $G_0$ and $W_2$ are expressed as:

$$G_0 = \frac{2e^2}{h}\cos^2\left(\frac{\pi m_d}{2}\right) \quad (4)$$

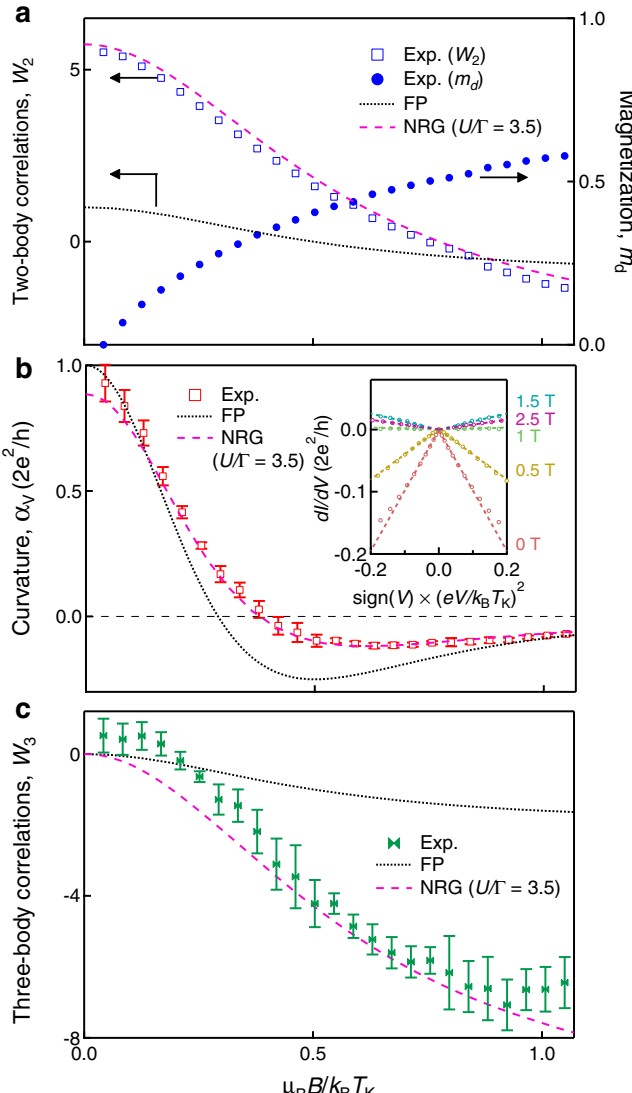

**Fig. 3 Two- and three-body correlations as a function of magnetic field.** **a** Two-body correlations $W_2$ and magnetization $m_d$ as a function of $\mu_B B/k_B T_K$. The scaling factor $k_B T_K$ is $138\mu$eV. $\mu_B B/k_B T_K = 1.0$ corresponds to $B = 2.4$T. The marks indicate the results deduced from the experiment. The dotted and dashed curves for $W_2$ are given by the free particle (FP) model and the NRG calculations ($U/\Gamma = 3.5$), respectively. **b** Curvature $\alpha_V$ as a function of $\mu_B B/k_B T_K$. The dotted and dashed curves are derived by the free particle (FP) model and the NRG calculations ($U/\Gamma = 3.5$), respectively. (Inset) $dI/dV$ as a function of sign$(V) \times (eV/k_B T_K)^2$ at different magnetic fields. The curves are offset for clarity. We obtained $\alpha_V$ with linear fitting. Error bars correspond to the uncertainty of the linear fit performed on slightly different ranges. **c** Three-body correlations $W_3$ as a function of $\mu_B B/k_B T_K$. In the analysis, we have compared the experimental results with the FP model (dotted curves) and the NRG results for the Anderson model with $U/\Gamma = 3.5$ (dashed curves)[20-23]. The error bars are determined based on those of $\alpha_V$ shown in Fig. 3b.

and

$$W_2 = \left[1 + 5(R-1)^2\right]\cos\left(\pi m_d\right), \tag{5}$$

respectively[20]. Note that to derive them we use the Friedel sum rule, $1 = \left(\delta_\uparrow + \delta_\downarrow\right)/\pi$, which holds even under the magnetic field. The gradual increase of $m_d$ deduced from the experimental $G_0$ is shown on the right axis in Fig. 3a. The numerical

renormalization group (NRG) calculations tell that the Wilson ratio is almost insensitive to the magnetic field up to $\mu_B B/k_B T_K \sim$ 1 (see Methods and Supplementary Note 2). Thus, as shown in Fig. 3a, the magnetic-field dependence of $W_2$ is directly deduced using the experimentally obtained $m_d$ and zero-field Wilson ratio $R = 1.95$[35]. To the best of our knowledge, such a quantitative measurement of the two-body correlation of the Kondo state under magnetic fields has never been reported.

Let us now analyze the non-equilibrium case of Fig. 2b in detail based on Eq. (1). The inset of Fig. 3b shows $dI/dV$ as a function of sign$(V) \times (eV/k_B T_K)^2$ at $B = 0, 0.5, 1, 1.5$, and 2.5T, respectively. The "V-shaped" curves indicate that the conductance shows a parabolic behavior around zero bias, and importantly, the sign of their curvature changes around 1T when $B$ increases. This sign reversal can also define the splitting of the Kondo peak, on which we focus here, instead of the above $2\Delta$[38]. We obtain the curvature, that is, the coefficient $\alpha_V$ from the fitting with Eq. (1) (see Methods and Supplementary Note 2). Figure 3b represents $\alpha_V$ as a function of the normalized magnetic field $\mu_B B/k_B T_K$. $\alpha_V$ crosses zero at $\mu_B B/k_B T_K = 0.38$ ($B = 0.9$T).

In order to elucidate the physical meaning of the magnetic field dependence of $\alpha_V$, we consider the FP model with a single resonant level (see Methods and Supplementary Note 1). In this case, analytic solutions of $\alpha_V$ are given only by using a single parameter, coupling constant ($\gamma_0$). $2\gamma_0$ corresponds to the half-width of $dI/dV$ at zero field. When $\gamma_0$ is given, the differential conductance in the magnetic field can be straightforwardly calculated for the FP case. $\alpha_V(B)$ thus obtained is shown in the dotted curve in Fig. 3b. Clearly, this dotted curve fails to explain the experimental observation, indicating the relevance of the Kondo correlation.

Now, using NRG calculations, we obtain $\alpha_V(B)$ for the Kondo-correlated QD with $U/\Gamma = 3.5$[20]. As shown in Fig. 3b, the theoretical curve (red dashed curve) satisfactorily reproduces the experiment.

Then what can we learn from this agreement between the experiment and the theory? As recently discussed[19-23], $\alpha_V$ is related to the three-body correlations $W_3$ as well as $W_2$:

$$\alpha_V(B) = \frac{2e^2}{h}\frac{\pi^2}{64}(W_2 + W_3) \times \left(\frac{T_K}{T_K^*}\right)^2. \tag{6}$$

Here, $W_3$ is defined as follows (see Methods and Supplementary Note 1),

$$W_3 = \frac{1}{2\chi_{\uparrow\uparrow}^2}\sum_\sigma \frac{\sin 2\delta_\sigma}{2\pi}\left(\chi_{\sigma\sigma\sigma} + 3\chi_{\sigma-\sigma-\sigma}\right). \tag{7}$$

As shown in Fig. 3a, we have already obtained the magnetic field dependence of $W_2$. In addition, $T_K^*(\varepsilon = 0, B)$ as a function of $B$ is obtained from the Lorentzian fitting of $dI/dV$ (see Supplementary Note 2). Thus, we can finally derive the magnetic-field dependence of $W_3$ using Eq. (6) as presented in Fig. 3c. $W_3$ is zero at zero field and becomes finite as $B$ is increased, i.e., as the TRS is broken.

We can also analytically derive $W_3$ in the FP model as shown in the dotted curve in Fig. 3c [see Methods and Supplementary Note 1]. Importantly, as is clear in this figure, the absolute value of the experimental $W_3$ is much larger than the theoretical value of the FP model. The behavior of $W_3$ is clearly due to the enhancement of the three-body correlations $\chi_{\sigma_1\sigma_2\sigma_3}$ by the Kondo effect. Indeed, NRG calculations with $U/\Gamma = 3.5$ shown in a red dashed curve nicely reproduce the experimental $W_3$. This result assures that the residual interaction manifests itself in the non-equilibrium transport in the symmetry-breaking regime. The experimental determination of the three-body correlations is our central achievement of this work.

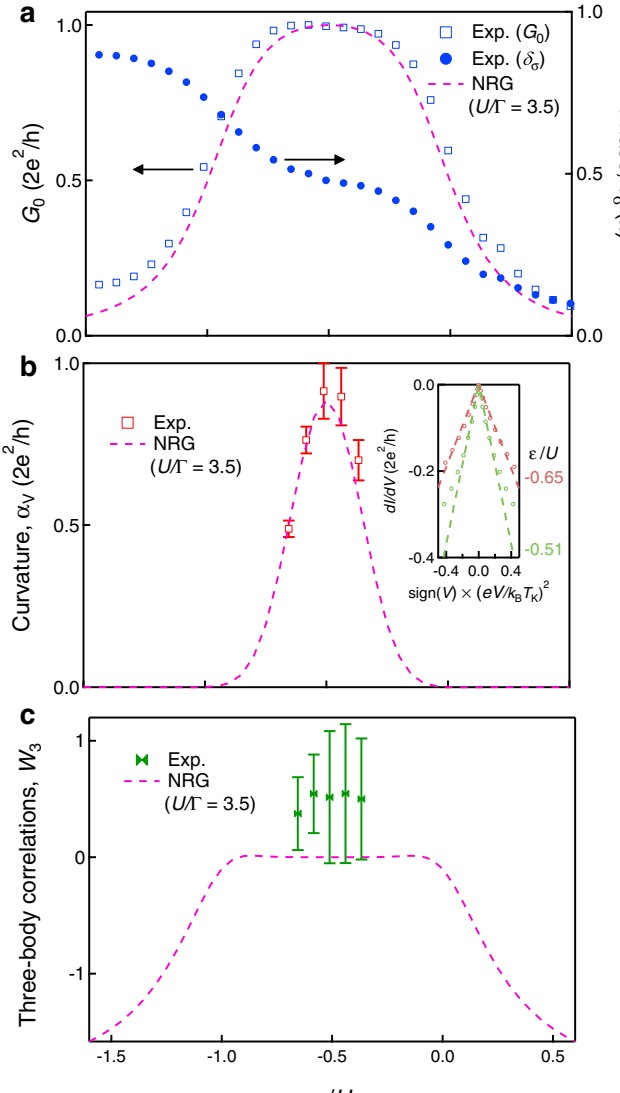

**Fig. 4 Two- and three-body correlations as a function of the energy level.** **a** Zero-bias conductance $G_0$ and phase shift $\delta_\sigma$ as a function of $\varepsilon/U$. The marks are experimental results obtained at $B = 0$. The dashed curve for $G_0$ is given by the NRG calculations with $U/\Gamma = 3.5$. **b** Curvature $\alpha_V$ as a function of $\varepsilon/U$. The dashed curve is the NRG results with $U/\Gamma = 3.5$. (Inset) $dI/dV$ as a function of $\text{sign}(V) \times (eV/k_B T_K)^2$ at $\varepsilon/U = -0.51$ and $-0.65$. Error bars correspond to the uncertainty of the linear fit performed on slightly different ranges. **c** Three-body correlations $W_3$ as a function of $\varepsilon/U$. We compare the experimental results with the NRG results with $U/\Gamma = 3.5$ (dashed curve)[20-23]. The error bars are determined based on those of $\alpha_V$ shown in Fig. 4b.

**Two- and three-body correlations in PHS broken case.** So far, we have focused on the PHS point ($\varepsilon/U = -0.5$, see Fig. 1c). Next, we address the PHS breaking regime by tuning the gate voltage ($\varepsilon$) at $B = 0$. The square mark in Fig. 4a shows $\varepsilon/U$ dependence of $G_0$, which decreases as $\varepsilon$ deviates from the PHS point. The dashed curve given by the NRG calculations again agrees with the experimental results well. We also show the phase shift, $\delta_\sigma (= \delta_\uparrow = \delta_\downarrow)$, evaluated from Eq. (1) (circles in Fig. 4a). As expected, the phase is locked around $\pi/2$ and develops a plateau[39,40]. The NRG calculations indicate that the Wilson ratio does not decrease in this region (see Supplementary Note 2). Relying on this fact, we evaluate the $\varepsilon/U$-dependence of the two-body correlation $W_2$ with Eq. (2) and the experimental value of

the Wilson ratio $R = 1.95$ at $B = 0$ (For the obtained $W_2$, see Supplementary Fig. 5).

The inset of Fig. 4b shows $dI/dV$ as a function of $\text{sign}(V) \times (eV/k_B T_K)^2$ at $\varepsilon/U = -0.51$ and $-0.65$. The slope at the former (around the PHS point) is larger than that at the latter (PHS broken). Figure 4b is the deduced curvature $\alpha_V$ as a function of $\varepsilon/U$. Here, we investigate the region within $|\varepsilon/U + 0.5| < 0.15$, where we experimentally obtain the Kondo temperature, $T_K^*(\varepsilon, B = 0 \text{ T})$ (see Fig. 2a). In Fig. 4c, we plot $W_3$ evaluated with Eq. (6), showing that $W_3$ remains almost zero in the entire region of $|\varepsilon/U + 0.5| < 0.15$, which is in sharp contrast with the above result in the magnetic field. This indicates that three-body correlations are more sensitive to magnetic field modulation than the gate voltage modulation (energy level modulation). This reflects that the spin degree of freedom of the Kondo state is fluctuating while the charge part is frozen. The NRG calculations with $U/\Gamma = 3.5$ shown in a dashed curve agree with the experimental $\alpha_V$ (Fig. 4b) and $W_3$ (Fig. 4c). This agreement again guarantees that the three-body correlations can be estimated experimentally.

## Discussion

In conclusion, we report the experimental determination of many-body correlations in the Kondo QD. We demonstrate that the curvatures of the differential conductance in the TRS- and PHS- broken regimes are well reproduced by the NRG calculations with strong interactions, enabling us to successfully evaluate the two-body and three-body correlations. The demonstrated quantitative analysis confirms the validity of the recent development of the Fermi liquid theory in the non-equilibrium regime, which should inspire future work, for example, to cover the temperature region around and above $T_K$. In particular, their sensitivity to TRS and PHS should make the three-body correlations an efficient tool to explore correlated quantum liquids such as topological spin liquids.

## Methods

**Fabrication process.** We show the outline of the fabrication process[41].

1. Sputtering Fe catalyst with 1 nm thickness on intrinsic Si substrate.
2. Placing the substrate in a quartz tube and heat it in a oven with low pressure ($\sim 1.1 \times 10^{-4}$ mbar) and stabilized temperature between 800 and 1000 °C.
3. Injecting $C_2H_2$ into the quartz tube for 9 seconds and pumping it out. After this process, we can get carbon nanotubes.
4. Connecting the carbon nanotubes with metallic leads by the electron beam lithography. We used Pd/Al bilayer for electrodes in the experiment [Supplementary Fig. 6a].

**Analysis in the $U = 0$ case.** The complete analytical form for Eq. (1) is obtained in the $U = 0$ case (see Supplementary Note 1), which is used in the FP model in this work. The experimental Kondo peak at zero field is treated as if it were a resonance peak with $U = 0$. We obtain $\gamma_0 = 65\,\mu eV$ by fitting the peak with the following formula:

$$\frac{dI}{dV} = \frac{2e^2}{h} \frac{\gamma_0^2}{(eV/2)^2 + \gamma_0^2}. \quad (8)$$

As $k_B T_K = 138\,\mu eV$, we see $k_B T_K/\gamma_0 \sim 2$, which is explained in Supplementary Note 1.

We present the magnetic field dependence of $W_2$, $\alpha_V$, and $W_3$ in Fig. 3a–c, respectively, by using:

$$W_2 = \frac{\gamma_0^2 - (\mu_B B)^2}{(\mu_B B)^2 + \gamma_0^2}, \quad (9)$$

$$\alpha_V = \frac{2e^2}{h} \frac{1 - 3(\mu_B B/\gamma_0)^2}{\{1 + (\mu_B B/\gamma_0)^2\}^3}, \quad (10)$$

$$W_3 = \frac{-2(\mu_B B)^2}{(\mu_B B)^2 + \gamma_0^2}. \quad (11)$$

We treat $\mu_B B/\gamma_0 = (k_B T_K/\gamma_0) \times (\mu_B B/k_B T_K)$. In Fig. 3a–c, $\mu_B B/k_B T_K$ is taken as the horizontal axis.

**Properties of $\chi_{\sigma_1 \sigma_2 \sigma_3}$.** Three-body correlations have permutation symmetry for the spin indexes:

$$\chi_{\sigma_1 \sigma_2 \sigma_3} = \chi_{\sigma_2 \sigma_3 \sigma_1} = \chi_{\sigma_3 \sigma_1 \sigma_2} = \chi_{\sigma_1 \sigma_3 \sigma_2} = \chi_{\sigma_2 \sigma_1 \sigma_3} = \chi_{\sigma_3 \sigma_2 \sigma_1}. \quad (12)$$

At zero magnetic field (the TRS point), $\chi_{\uparrow\uparrow\uparrow} = \chi_{\downarrow\downarrow\downarrow}$ and $\chi_{\uparrow\downarrow\downarrow} = \chi_{\uparrow\uparrow\downarrow}$ hold. At the PHS point, $\chi_{\uparrow\uparrow\uparrow} = -\chi_{\downarrow\downarrow\downarrow}$ and $\chi_{\uparrow\downarrow\downarrow} = -\chi_{\uparrow\uparrow\downarrow}$ hold. These properties are used in the discussion of $W_3$ defined by Eq. (7).

**Analysis of differential conductance.** In the analysis shown in Fig. 2b, we do not take the data obtained at very low bias region (typically $|eV/k_B T_K|^2 < 0.01$), where a slight deviation from the parabolicity is observed. This effect is most probably due to some other lower energy physics such as two-stage Kondo effect (see Supplementary Note 2 and ref. [42]).

**Numerical calculations.** NRG results shown in Figs. 3 and 4 have been calculated using the Anderson impurity model in the form $\mathcal{H} = \mathcal{H}_d + \mathcal{H}_c + \mathcal{H}_T$,

$$\mathcal{H}_d = \sum_{\sigma=\uparrow,\downarrow} \varepsilon_\sigma \, d_\sigma^\dagger d_\sigma \; + \; U \, d_\uparrow^\dagger d_\uparrow \, d_\downarrow^\dagger d_\downarrow,$$

$$\mathcal{H}_c = \sum_{\sigma=\uparrow,\downarrow} \sum_{\lambda=L,R} \int_{-D}^{D} d\xi \; \xi \; c_{\xi\lambda\sigma}^\dagger c_{\xi\lambda\sigma},$$

$$\mathcal{H}_T = \sum_{\sigma=\uparrow,\downarrow} \sum_{\lambda=L,R} v_\lambda \int_{-D}^{D} d\xi \; \sqrt{\rho_c} \left( c_{\xi\lambda\sigma}^\dagger d_\sigma + \text{H.c.} \right).$$

Here, $\varepsilon_\sigma = \varepsilon - \sigma\mu_B B$ is the energy of a single electron with spin $\sigma$ and $U$ is the Coulomb interaction between electrons in the QD. The g-factor is taken as 2. $\mathcal{H}_c$ describes the conduction bands of the leads on the left ($L$) and right ($R$) with the constant density of states $\rho_c \equiv 1/(2D)$. When $U = 0$, the resonance width of the local level due to the tunnel couplings in $\mathcal{H}_T$ is given by $\Gamma/2 = \pi\rho_c(v_L^2 + v_R^2)$. We assume that the couplings to be symmetric $v_L = v_R$.

The NRG calculations have been carried out choosing the discretization parameter to be $\Lambda = 2.0$ and keeping $N_{\text{trunc}} = 3600$ low-lying energy states at each step of the iterative procedure[43]. Our NRG code uses the global $U(1) \otimes SU(2)$ symmetries and the method for deducing the correlation functions described in ref. [23].

## Data availability

The data that support the findings of this study are available from the corresponding author upon reasonable request.

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

## Acknowledgements

We thank Raphaëlle Delagrange for helpful discussions and for sample fabrication. K.K. acknowledges the stimulating discussion in the meeting of the Cooperative Research Project of the RIEC, Tohoku University. This work was partially supported by JSPS KAKENHI Grant (No. JP19K14630, No. JP19H05826, No. JP19H00656, No. JP18K03495, JP18H01815, and No. JP18J10205), JST CREST Grant (No. JPMJCR1876), and the French program ANR JETS (ANR-16-CE30-0029-01).

## Author contributions

M.F. and R.D. fabricated the sample. T.H., M.F., T.A., and S.L. performed the experiments. T.H. analyzed the data. Y.T., R.S., and A.O. performed the theoretical simulations. T.H. and K.K. wrote the paper with the input of all authors. K.K. and A.O. supervised the research.

## Competing interests

The authors declare no competing interests.
