## [Peer Review File · Nature Communications]

REVIEWER COMMENTS

Reviewer #1 (Remarks to the Author):

Referee Report to "Three-body correlations in nonlinear response of correlated quantum liquid"

In this paper, the authors report clear experimental signatures of interaction effects in the non-linear corrections to the differential conductance of a quantum dot device composed of a carbon nanotube connected to metallic leads. The interest of the reported experiment resides in the fact that it clearly probes the leading out-of-equilibrium properties of a strongly correlated system in the Kondo regime. Additionally, it provides the first experimental demonstration of the validity of effective Fermi-liquid approaches, which have been recently devised to describe such out-of-equilibrium corrections. These approaches highlighted the importance of interactions and three-body correlations for out-of-equilibrium these corrections.

The quality of the data and their presentation [including consistency checks and the comparison with the free particle model] is excellent. The agreement with the theoretical predictions is also impressive and the paper is clearly written. The realization of such a high-quality experiment will motivate theorists and experimentalists in the condensed matter community to address important open questions in the out-of-equilibrium dynamics of strongly correlated systems, as it will appear in a question below. For all these reasons, I believe that this work deserves to be published in Nature Communications.

Honestly, I do not have any major criticisms concerning the paper, apart the small comments reported below.

- In the Introduction, it is not clear what the authors mean exactly with two-body correlations $\chi_{\{\sigma_1\sigma_2\}}$, the definition of this quantity should be given immediately.

- caption Fig. 1 : applied to QD -> applied to the QD

- Page 2. : I have found the sentence "In the $U = 0$ case, the complete analytical form for Eq. (1) is obtained, where U is the charging energy of QD (see Supplementary Information)" a little bit

misleading. It is not maybe clear to the reader what would be the meaning of T_k in the $U \rightarrow 0$ limit. It would be maybe useful to include some information from the supplementary material.

- Did the authors perform measurements for the non-linear corrections to the conductance at temperatures T of the order of T_K ? This is a question of fundamental interest. In this regime, effective Fermi-liquid approach would fail and scarce reliable theoretical results and understanding are available in the literature. An experimental setup of the quality presented in this work, would allow to clarify the role of the interactions in the crossover leading to the Kondo regime. The authors should maybe comment on this issue in the conclusion of the paper.

Reviewer #2 (Remarks to the Author):

This is a combined experimental/theoretical work with a strong emphasis on theory. It targets the nonlinear response of the Fermi liquid given by the Kondo effect in a carbon nanotube. Based on Fermi liquid theory, the authors derive the strength of three-body correlations with relation to the distance in parameter space from particle-hole symmetry and time-reversal symmetry. The nonlinear response of a Kondo system is a highly interesting and complex topic and has been under debate for long time. As such, the presented results are definitely worth publication.

* Unpacking the manuscript took some effort. I would like to suggest improving Figure 1(a), which I found quite confusing, however, unfortunately I do not have a clear idea on how to precisely do that. In the case of two-particle correlations, there are two obvious distinct cases - equal and opposite spin - which fulfil different functions, see the formulas for T_k and the Wilson ratio. Can any similar classification be done for three particles? (In any case this is only a suggestion.)

A carbon nanotube, due to its inherent spin *and* valley degeneracy, can display both the SU(2) and the SU(4) Kondo effect.

* The discussion does not state anywhere (or I can't find it) which case is discussed. The SU(2) case is implicit in that the theory talks only about a single spin-1/2 system, but this should in any case be clarified in the text and maybe even in the abstract / introductory paragraph.

* How do the authors justify that their data is described by the SU(2) Kondo effect, when also SU(4) can occur in carbon nanotubes? The theory obviously "fits" (and the data also looks like SU(2)), however, this is a gap in the argumentation. And in particular when the fourfold degeneracy is not strongly enough broken the additional states have a clear impact on the nonlinear conductance.

Dimensionless axis labels (e.g., $eV/k_B T_K$) are very appealing from a theoretical point of view but make a comparison of experimental data difficult. May I suggest adding the scaling factors (as in this particular case $k_B T_K$ in eV) in the figure captions for readability?

As a side note, reaching the unitary limit does not require the nanotube to be particularly "clean". As the authors are certainly aware, it mostly means that it couples equally well to both contacts.

Finally, given that it is also an experimental work and extensively discusses few data curves of one(?) device, we should probably get to know the device a bit more. (This should be in the supplement, and referred to in the main text. Given the nice quality of the already shown data in combination with the universal behaviour of the Kondo effect, I don't expect this to have any impact on the statement and theoretical outcome, but it would be useful to place the findings into context.)

I would expect at least

- * a brief discussion of device fabrication, materials and geometry
- * a "Coulomb diamonds" style plot of the differential conductance at the Kondo ridge of Fig. 2 and Fig. 3 and its surroundings
- * possibly similar data in a magnetic field as point of reference for Fig. 3

If the device is identical to one shown in referenced publications (Ref. 33?) and this information is already shown elsewhere, please state so explicitly! Then again, space in the supplement is cheap, and it may be useful to have reference data in place.

Reply to Referee #1

We thank the referee very much for recommending our work to be published in Nature Communications and noting that “The quality of the data and their presentation [including consistency checks and the comparison with the free particle model] is excellent.” We are very happy to have such high evaluation. We also highly appreciate several precious comments to further improve our manuscript.

In the following, we would like to answer his/her comments and questions. Please also see the list of the revisions below and the color parts in the main text. In order to follow the formatting instructions of Nature Communications, we have added section names and figure titles, made many minor corrections in all the sections, and moved several explanations to Methods. These changes are not highlighted to make the manuscript easy to read. (For clarity, your comments are shown in *blue and italic.*)

[Referee’s comment (1)]

In the Introduction, it is not clear what the authors mean exactly with two-body correlations $\chi_{\sigma_1\sigma_2}$, the definition of this quantity should be given immediately.

[Answer]

As the referee suggested, we have added the definition of two-body correlation immediately after the explanation of $\chi_{\sigma_1\sigma_2}$ in the 3rd paragraph. The two-body correlation is exactly conventional linear susceptibility.

[Referee’s comment (2)]

caption Fig. 1 : applied to QD -> applied to the QD

[Answer]

As the referee pointed out, we have included the term, “the”, at the 2nd to last sentence of caption of Fig. 1c.

[Referee’s comment (3)]

Page 2. : I have found the sentence "In the $U = 0$ case, the complete analytical form for Eq. (1) is obtained, where U is the charging energy of QD (see Supplementary Information)" a little bit misleading. It is not maybe clear to the reader what would be the meaning of T_k in the $U \rightarrow 0$ limit. I would be maybe useful to include some information

from the supplementary material.

[Answer]

In order to avoid readers' misunderstanding of the 1st sentence in the 7th paragraph, we have changed the sentence to "Importantly, when the onsite Coulomb interaction (U) in the QD is zero, a complete analytical form for Equation (1) is obtained by replacing $k_B T_K$ with $2\gamma_0$ (see Methods and Supplementary Note 1). Here, $2\gamma_0$ virtually corresponds to the half width of a resonance peak." We have also shown the detail analysis procedure for $U = 0$ case in the subsection "Analysis in the $U = 0$ " in Methods.

[Referee's comment (4)]

Did the authors perform measurements for the non-linear corrections to the conductance at temperatures T of the order of T_K ? This is a question of fundamental interest. In this regime, effective Fermi-liquid approach would fail and scarce reliable theoretical results and understanding are available in the literature. An experimental setup of the quality presented in this work, would allow to clarify the role of the interactions in the crossover leading to the Kondo regime. The authors should maybe comment on this issue in the conclusion of the paper.

[Answer]

We thank you for your nice comment, which is certainly an important point. Unfortunately, we do not access the regime at temperature of the order of T_K . The reason is that $T_K \sim 1.6$ K is higher than the highest temperature for stable operation of our dilution fridge (800mK), and systematic measurement around and above T_K was not easy. We have added comments on this issue in Discussion, as the referee suggested.

Finally, we thank the referee very much again for his/her fruitful comments. We strongly believe that the manuscript has been further improved thanks to them and would be very happy that he/she would find our manuscript now publishable.

Sincerely yours,

Tokuro Hata and Kensuke Kobayashi

Tokuro Hata

Department of Physics, Graduate School of Science, Osaka University
1-1 Machikaneyama, Toyonaka, Osaka 560-0043, Japan

TEL: +81-3-5734-2809 e-mail: hata@phys.titech.ac.jp

Kensuke Kobayashi

Department of Physics, Graduate School of Science, Osaka University

1-1 Machikaneyama, Toyonaka, Osaka 560-0043, Japan

Institute for Physics of Intelligence and Department of Physics, The University of Tokyo

7-3-1 Hongo, Bunkyo-ku, Tokyo 113-0033

TEL: +81-3-5841-4126 e-mail: kensuke@phys.s.u-tokyo.ac.jp

Reply to the Referee #2

We thank the referee very much for recommending our work to be published in Nature Communications and noting that “the presented results are definitely worth publication.” We also highly appreciate several precious comments to further improve our manuscript.

In the following, we would like to answer his/her comments and questions. Please also see the list of the revisions below and the color parts in the main text. In order to follow the formatting instructions of Nature Communications, we have added section names and figure titles, made many minor corrections in all the sections, and moved several explanations to Methods. These changes are not highlighted to make the manuscript easy to read. (For clarity, your comments are shown in *blue and italic.*)

[Referee’s comment (1)]

Unpacking the manuscript took some effort. I would like to suggest improving Figure 1(a), which I found quite confusing, however, unfortunately I do not have a clear idea on how to precisely do that. In the case of two-particle correlations, there are two obvious distinct cases - equal and opposite spin - which fulfil different functions, see the formulas for T_k and the Wilson ratio. Can any similar classification be done for three particles? (In any case this is only a suggestion.)

[Answer]

For readers to understand three-body correlations, we have added a schematic view of the correlation between the three electrons accounting for the Fermi liquid correction as Fig. 1a. We hope such a simplified picture would help readers to have some intuition on the three-body correlation. In addition, in the 6th paragraph, we have mentioned that two independent parameters, $\chi_{\uparrow\uparrow}$ and $\chi_{\uparrow\downarrow}$, characterize the nonlinear response for specific cases shown in the subsection “Properties of $\chi_{\sigma_1\sigma_2\sigma_3}$ ” in Methods.

In addition, we take your comment “*Unpacking the manuscript took some effort.*” seriously, and we have made several minor revisions to enhance the readability such as by adding mutual references to Figures and Equations in the manuscript. We thank you very much for your helpful comment.

[Referee’s comment (2)]

*A carbon nanotube, due to its inherent spin *and* valley degeneracy, can display both*

the SU(2) and the SU(4) Kondo effect.

** The discussion does not state anywhere (or I can't find it) which case is discussed. The SU(2) case is implicit in that the theory talks only about a single spin-1/2 system, but this should in any case be clarified in the text and maybe even in the abstract / introductory paragraph.*

** How do the authors justify that their data is described by the SU(2) Kondo effect, when also SU(4) can occur in carbon nanotubes? The theory obviously "fits" (and the data also looks like SU(2)), however, this is a gap in the argumentation. And in particular when the fourfold degeneracy is not strongly enough broken the additional states have a clear impact on the nonlinear conductance.*

[Answer]

We thank you for pointing out this important point, which we failed to mention in the original manuscript. As you wrote, we discuss only the SU(2) Kondo state in this manuscript. We have revised the manuscript to explicitly mention that we are dealing with the SU(2) Kondo case. In the 9th paragraph, we write that the symmetry of the Kondo state was experimentally identified by the shape of Coulomb diamond and shot noise measurement as demonstrated before [Ferrier et al, Nat Phys. 12, 230 (2016); Phys. Rev. Lett. 118, 196803 (2017)]. The effective charge of the shot noise is clearly different between in the SU(2) case and in the SU(4) case and thus we are perfectly sure that the SU(2) Kondo state is realized in this manuscript.

[Referee's comment (3)]

Dimensionless axis labels (e.g., $eV/k_B T_K$) are very appealing from a theoretical point of view but make a comparison of experimental data difficult. May I suggest adding the scaling factors (as in this particular case $k_B T_K$ in eV) in the figure captions for readability?

[Answer]

We thank you for this nice comment to increase the readability. As the referee suggested, we added the scaling factor, $k_B T_K = 138 \mu\text{eV}$ and have mentioned that $\mu_B B / k_B T_K = 1.0$ corresponds to $B = 2.4 \text{ T}$ in the caption of Fig. 3b.

[Referee's comment (4)]

As a side note, reaching the unitary limit does not require the nanotube to be particularly "clean". As the authors are certainly aware, it mostly means that it couples equally well

to both contacts.

[Answer]

As the referee pointed out, the cleanness is not required for achieving the unitary limit. We removed the term “clean”.

[Referee’s comment (5)]

Finally, given that it is also an experimental work and extensively discusses few data curves of one(?) device, we should probably get to know the device a bit more. (This should be in the supplement, and referred to in the main text. Given the nice quality of the already shown data in combination with the universal behaviour of the Kondo effect, I dont expect this to have any impact on the statement and theoretical outcome, but it would be useful to place the findings into context.)

I would expect at least

- * a brief discussion of device fabrication, materials and geometry*
- * a "Coulomb diamonds" style plot of the differential conductance at the Kondo ridge of Fig. 2 and Fig. 3 and its surroundings*
- * possibly similar data in a magnetic field as point of reference for Fig. 3*

If the device is identical to one shown in referenced publications (Ref. 33?) and this information is already shown elsewhere, please state so explicitly! Then again, space in the supplement is cheap, and it may be useful to have reference data in place.

[Answer]

As the referee pointed out, the device is identical to the one studied in Ref. [35, 36, 37], which we explicitly mentioned in the 9th paragraph. We also added a section to show the fabrication process and supplemental data in the Supplementary Note 4.

Finally, we thank the referee very much again for his/her fruitful comments. We strongly believe that the manuscript has been further improved thanks to them and would be very happy that he/she would find our manuscript now publishable.

Sincerely yours,

Tokuro Hata and Kensuke Kobayashi

Tokuro Hata

Department of Physics, Graduate School of Science, Osaka University
1-1 Machikaneyama, Toyonaka, Osaka 560-0043, Japan
TEL: +81-3-5734-2809 e-mail: hata@phys.titech.ac.jp

Kensuke Kobayashi
Department of Physics, Graduate School of Science, Osaka University
1-1 Machikaneyama, Toyonaka, Osaka 560-0043, Japan

Institute for Physics of Intelligence and Department of Physics, The University of Tokyo
7-3-1 Hongo, Bunkyo-ku, Tokyo 113-0033
TEL: +81-3-5841-4126 e-mail: kensuke@phys.s.u-tokyo.ac.jp

List of the revisions

According to Comment (1) from Referee #1, we have added “Here, the two-body correlation is identical to the conventional linear susceptibility, which is defined as the differential of the free energy Ω of the total system regarding the energies of two electrons, $\chi_{\sigma_1\sigma_2} \equiv -\partial^2\Omega/\partial\varepsilon_{\sigma_1}\varepsilon_{\sigma_2}$ (see Supplementary Note 1).” in 3rd paragraph [in the right column of p.1 of the highlighted pdf].

According to Comment (2) from Referee #1, we have changed “applied to QD” to “applied to the QD” in the second to last sentence in the caption of Fig. 1c.

According to Comment (3) from Referee #1, we have modified the sentence, “In the $U = 0$ case, the complete analytical form for Eq. (1) is obtained, where U is the charging energy of QD (see Supplementary Information).” to “Importantly, when the onsite Coulomb interaction (U) in the QD is zero, a complete analytical form for Equation (1) is obtained by replacing $k_B T_K$ with $2\gamma_0$ (see Methods and Supplementary Note 1). Here, $2\gamma_0$ virtually corresponds to the half width of a resonance peak.” [in the right column of p.2 of the highlighted pdf]. We have also shown the detail analysis procedure for $U = 0$ case in the subsection “Analysis in the $U = 0$ ” in Methods.

According to Comment (4) from Referee #1, we have added “The demonstrated quantitative analysis confirms the validity of the recent development of the Fermi liquid theory in the non-equilibrium regime, which should inspire future work, for example, to cover the temperature region around and above T_K .” in Discussion [in the left column of p.6 of the highlighted pdf].

According to Comment (1) from Referee #2, we have added a schematic view of the correlation as Fig. 1a.

According to Comment (1) from Referee #2, we have changed “ $\chi_{\sigma_1\sigma_2\sigma_3}$ ” to “ $\chi_{\uparrow\uparrow}$ and $\chi_{\uparrow\downarrow}$ for specific cases” in the 3rd to last sentence in the 6th paragraph [in the right column of p.2 of the highlighted pdf]. The specific cases are shown in the subsection “Properties $\chi_{\sigma_1\sigma_2\sigma_3}$ ” in Methods.

According to Comment (2) and (5) from Referee #2, we have added the sentences “We focus on the conventional SU(2) Kondo state previously reported in Refs. [35-37]. The details are presented in those references and in Supplementary Note 4. The shape of the Coulomb diamond and the shot noise measurement experimentally determines whether the symmetry of the Kondo state is SU(2) or SU(4) [35-37].” in the 9th paragraph [in the left column of p.3 of the highlighted pdf].

According to Comment (3) from Referee #2, we have added the sentence “The scaling factor $k_B T_K$ is 138 μeV . $\mu_B B / k_B T_K = 1.0$ corresponds to $B = 2.4$ T.” in the caption of Fig. 3b.

According to Comment (4) from Referee #2, we have removed “clean” from the last sentence in 6th paragraph [in the right column of p.2 of the highlighted pdf].

According to Comment (5) from Referee #2, we have added a section, “Supplementary Note 4: Device” in Supplementary Information, where the fabrication process and supplemental data are shown.

We have added “ $1 = (\delta_\uparrow + \delta_\downarrow) / \pi$ ” in the 12th paragraph. [in the right column of p.4 of the highlighted pdf].

We have replaced Γ_0 with γ_0 in 7th and 14th paragraph [in the right column of p.4 of the highlighted pdf].

We have added a legend in Fig. 3a to indicate experimental points of W_2 .

We have added the subsection “Numerical calculations” in Methods.

We have added “Error bars correspond to the uncertainty of the linear fit performed on slightly different ranges.” in the caption of Fig. 3b.

After having submitted this manuscript, two theory papers written by several authors of this work were published. As both are very closely related to this work, we cited them in the revised version as Refs. [22, 23].

According to Referee #2's comment "*Unpacking the manuscript took some effort.*", we have made several minor revisions to enhance the readability such as by adding mutual references to Figures and Equations in the manuscript. Also, several minor usages and typos are corrected.

In the original manuscript, we used the Wilson ratio $R = 2.0$ to obtain Figs. 3a and c. In the revised manuscript, to be more exact, we show Figs. 3a and c deduced based on the experimental value $R = 1.95$. Accordingly, we have changed the sentence " $W_2 = 6.0$ is obtained at the symmetric point ($\delta_\sigma = \pi/2$)."

 to " $W_2 = 5.5$ at $R = 1.95$ and the symmetric point ($\delta_\sigma = \pi/2$)." These are very minor modifications, and the plots have changed only slightly. Thus, these do not at all affect our discussion.

According to comments from the editor, we have changed the format of the manuscript as follows:

1. We have added abstract.
2. We have made "Introduction" and "Discussion" without subheading, and "Results" sections with subheadings, "Nonlinear conductance in Kondo regime", "Sample and measurement setup", "Basic characteristics of the Kondo state", "Two- and three-body correlation in TRS broken case", and "Two- and three-body correlation in PHS broken case". Accordingly, we have many minor corrections to the manuscript.
3. We have made "Methods" section, which consists of "Analysis in the $U = 0$ ", "Properties of $\chi_{\sigma_1\sigma_2\sigma_3}$ ", and "Numerical calculations".
4. We have added short title for each figure.
5. We have replaced the general citations to the Supplementary Information with specific citations such as "see Supplementary Note 1".
6. We have removed parentheses for figure labels, a, b, and c such as the change, "Fig. 1(a)" \rightarrow "Fig. 1a". Accordingly, we have replaced brackets for additional explanation with parentheses such as the change, "[see Fig. 1(c)]" \rightarrow "(see Fig. 1c)".

REVIEWERS' COMMENTS

Reviewer #1 (Remarks to the Author):

I am satisfied with the answers given by the authors and I confirm my positive evaluation of this work to be published in Nature Communications.

Reviewer #2 (Remarks to the Author):

I am fully satisfied with the revision, and would like to thank the authors for the beautiful work. The paper should be published as-is now.